# MicroRNA193a: An Emerging Mediator of Glomerular Diseases

**DOI:** 10.3390/biom13121743

**Published:** 2023-12-04

**Authors:** Joyita Bharati, Megan Kumar, Neil Kumar, Ashwani Malhotra, Pravin C. Singhal

**Affiliations:** 1Feinstein Institute for Medical Research, Manhasset, NY 11030, USA; sharma.joyita4@gmail.com (J.B.); mkumar24@lawrenceville.org (M.K.); neilkumar@chicago.edu (N.K.);; 2Division of Kidney Diseases and Hypertension, Zucker School of Medicine at Hofstra Northwell Health, Great Neck, NY 11021, USA

**Keywords:** miRNA, miR-193a, FSGS, diabetes mellitus, membranous glomerulopathy, biomarkers

## Abstract

MicroRNAs (miRNAs) are noncoding small RNAs that regulate the protein expression of coding messenger RNAs. They are used as biomarkers to aid in diagnosing, prognosticating, and surveillance of diseases, especially solid cancers. MiR-193a was shown to be directly pathogenic in an experimental mouse model of focal segmental glomerulosclerosis (FSGS) during the last decade. Its specific binding and downregulation of Wilm’s tumor-1 (WT-1), a transcription factor regulating podocyte phenotype, is documented. Also, miR-193a is a regulator switch causing the transdifferentiation of glomerular parietal epithelial cells to a podocyte phenotype in in vitro study. Interaction between miR-193a and apolipoprotein 1 (APOL1) mRNA in glomeruli (filtration units of kidneys) is potentially involved in the pathogenesis of common glomerular diseases. Since the last decade, there has been an increasing interest in the role of miR-193a in glomerular diseases, including diabetic nephropathy and membranous nephropathy, besides FSGS. Considering the lack of biomarkers to manage FSGS and diabetic nephropathy clinically, it is worthwhile to invest in evaluating miR-193a in the pathogenesis of these diseases. What causes the upregulation of miR-193a in FSGS and how the mechanism is different in different glomerular disorders still need to be elucidated. This narrative review highlights the pathogenic mechanisms of miR-193a elevation in various glomerular diseases and its potential use in clinical management.

## 1. Introduction

Glomerular diseases, including primary and secondary causes like diabetes, are the most common cause of end-stage kidney disease. End-stage kidney disease (ESKD) is characterized by the loss of kidney function—a glomerular filtration rate <15 mL/min or less requiring dialysis or kidney transplantation. ESKD incidence has increased steadily worldwide [1]. Most glomerular diseases involve podocyte injury with variable involvement of other components of glomeruli (Figure 1); some start with podocyte injury, and others result in podocyte injury. Further, progress in understanding podocyte biology has been rapid, and newer players are increasingly identified. MicroRNAs (miRNAs) are involved in critical cellular processes in developing and maintaining the structure of resident kidney cells [2] (Figure 1) and are essential regulators of disease pathways. miR-193a is an example that was proven to directly cause FSGS in an experimental model [3]. Following the initial publication, there has been an explosive interest in this molecule among researchers working on kidney diseases. This narrative review discusses the role of miR-193a in different glomerular diseases and its potential use in the clinical management of these diseases.

## 2. What Are microRNAs?

MicroRNAs (miRNAs) are noncoding small regulatory RNAs (18–22 nucleotides) that act as post-transcriptional repressors of target messenger RNAs (mRNAs). They were discovered in the 1990s [4], and approximately 2600 microRNAs are observed to regulate 60% of human protein-coding genes [5]. Like typical RNA, miRNAs are formed in the nucleus as primary miRNAs (pri-miRNA), a stem–loop structure containing two double-stranded RNAs. Pri-miRNA is cleaved into precursor miRNA (pre-miRNA), a stem–loop structure containing one double-stranded RNA, by an RNase III enzyme called Drosha. Pre-miRNA is exported to the cytoplasm, where it is cleaved into a double-stranded miRNA by another RNase III enzyme called Dicer, which gets incorporated into the RNA-induced silencing complex. Here, one strand of the double-stranded miRNA is degraded, making only one strand free to bind the target mRNA. The miRNAs are named miR-5p or miR-3p based on the original strand of pre-miRNA from which they are derived. The 5’ region of the miRNA (miR-5p pr miR-3p) binds to the 3’ untranslated region of the target mRNA to repress it [6]. miRNAs are transported in body fluids inside exosomes. Exosomes are formed when multivesicular bodies inside cells (parts of endosomes) [7] fuse with the plasma membrane and are shed [8]. Exosomes contain a unique pattern of RNAs, DNAs, and miRNAs based on their cellular origin. Most cells in the human body can secrete exosomes, and it is apparent now that exosomes are a way of intercellular communication. Exosomes carrying miRNAs are used as biomarkers of various types of cancers, as specific tumor cells are observed to release exosomes containing unique profiles of miRNAs [9]. Exosomes carrying individual miRNA profiles in the blood and urine are being investigated in glomerular diseases [10].

One type of microRNA can regulate 100 different protein-coding genes, and different microRNAs can target a single protein-coding gene. They are crucial in biological pathways such as cell proliferation, differentiation, apoptosis, tumorigenesis, and fibrosis. miRNAs are involved in kidney development and diseases such as allograft rejection, acute kidney injury, and chronic kidney disease, including glomerulonephritis [2]. Different types of miRNAs can have a contradictory effect on different cells, e.g., miR-21 has an antiapoptotic effect on tubular cells, whereas miR-193a is proapoptotic for podocytes. miR-30 and miR-146a are anti-inflammatory and antifibrotic, whereas miR92a and miR-193a are profibrotic. miR-193a increases cell proliferation and migration in renal cell carcinoma [11], osteosarcoma, and pancreatic carcinoma [12]; however, it has tumor-suppressive effects in other cancers, such as colorectal, breast, and gastric cancer [13]. The upregulation of specific miRNAs and the downregulation of other specific miRNAs are associated with diseases and are determined by genetic and epigenetic changes [12].

## 3. miRNAs in Glomerular Diseases

Glomerular diseases are characterized by pathological alteration of the glomerulus with distinct patterns in different glomerular diseases based on the predominant glomerular cell/structure involved and pathogenesis. Most glomerular diseases are clinically characterized by albuminuria with or without progressive kidney dysfunction. Although often used interchangeably, *glomerulonephritis* (*GN*) represents a type of *glomerular disease* characterized by inflammatory cells within the glomeruli. Focal segmental glomerulosclerosis (FSGS), diabetic nephropathy, and membranous nephropathy are common causes of glomerular diseases that lack typical inflammation in glomeruli, with the podocyte being the dominant cell injured. The incidence and prevalence of FSGS have increased over the last two to three decades [14]; its pathogenesis is still not understood. The classic albuminuric type of diabetic nephropathy is characterized by a predominant involvement of the glomerulus, starting with podocyte injury from various downstream effects of hyperglycemia. Membranous nephropathy is characterized by antibodies targeting specific podocyte antigens, causing immune-complex deposition in the glomerular filtration barrier with resultant damage [15].

Around 700 miRNAs are known to be expressed in normal human kidneys. The deletion of podocyte-specific Dicer, an RNAase III enzyme required for generating microRNA, in experimental mice has been shown to lead to albuminuric kidney disease and death by six weeks of age [16]. Dicer deletion inhibited miR-30, increasing target genes, including receptors for advanced glycation end product, vimentin, heat-shock protein 20, and immediate early response [3,17]. miR-30, highly expressed in glomeruli, protects podocytes by decreasing p53 and Notch-1 expression (mediators of podocyte injury) [18]. Interestingly, pretreatment with glucocorticoids preserved miR-30 expression in cultured podocytes despite their treatment with puromycin aminonucleoside and lipopolysaccharide (podocyte cytotoxic) [18]. A combination of high miR-196a, high miR-30a, and high miR-490 in the urine predicted active FSGS, diabetic nephropathy, and membranous nephropathy (MN) [19]. However, these three miRNAs were similar in active and remitted diabetic nephropathy and MN, unlike FSGS.

Specific miRNAs are involved in glomerular diseases like diabetic nephropathy, lupus nephritis, IgA nephropathy, FSGS [2,6], antineutrophilic cytoplasmic antibody (ANCA)-associated vasculitis [20], and membranous nephropathy [21]. Plasma miR-125b-5p and miR-186-50 are observed to be associated with FSGS disease activity.

miR-93 promotes glomerular injury through vascular endothelial growth factors. miR-195 activates apoptosis in podocytes by inhibiting Bcl-2. Collagen deposition in the mesangium and interstitial fibrosis in diabetic nephropathy is promoted by miR-192. miR-135 is implicated in activating the Wnt/beta-catenin pathway in glomerular diseases like FSGS and diabetic nephropathy [22], where this pathway contributes to podocyte injury by increasing Snail expression, a transcriptional regulator [23]. These studies affirm the critical role of miRNAs in podocyte biology and their potential utility in the clinical management of glomerular diseases, partly because podocytes are regulated indirectly by microRNAs. Interestingly, some microRNAs are directly implicated in causing kidney injury [3,24], and others protect against injury [18,25]. Table 1 summarizes miRNAs involved in different glomerular diseases.

## 4. Why Study microRNA?

MicroRNAs are stable-in-body fluids such as plasma and urine in exosomes or vesicles. This makes them versatile as noninvasive biomarkers to diagnose diseases as they can be easily measured. miRNAs are secreted by cells in exosome-encapsulated form and can target other types of cells locally and distantly. Urinary miRNAs are either passively filtered by glomeruli or secreted by tubules.The role of a few miRNAs is well studied in the pathogenesis of human diseases. Specific miRNAs have been known to be involved in kidney disease for the last two decades. Therefore, miRNAs can be used as diagnostic, prognostic, surveillance, and predictive biomarkers in many human diseases.Chemical modification of miRNAs (oligonucleotide inhibitors), tandem repeats of miR-binding sites (decoy or sponge), and inhibition of miRNAs using nanoparticles/nucleic acids (anti-miRs) can be delivered to cells, and they can unmask the effects of miRNAs under different conditions. Therefore, miRNAs can be developed as therapeutic agents and could act as therapeutic targets as well.

## 5. microRNA-193a in Glomerular Diseases

miR-193a-3p is predominantly expressed by glomerular parietal epithelial cells (PECs) amongst kidney cells besides almost all other organ cells in the human body except the bladder, cervix, and brain parts [13]. Adipose tissue and breast tissue have the highest miR-193a-3p expression under normal conditions. The existing literature suggests a role of miR-193a-3p in tumor suppression as a promoter of tumor cell apoptosis, mainly via inhibition of cell cycle (G1-S transition) in breast cancer and glioblastoma cell lines [13]. Interestingly, research on miR-193a’s association with kidney disease started after observing the rapid development of fatal FSGS in transgenic mice with mammary tumors induced with miR-193a [3]. The regulatory effect of miR-193a on podocytes (not PECs) is similar to its effect on tumor cells, mainly via WT-1 expression. In vitro studies suggest that miR-193a binds to WT-1 mRNA, preventing its expression. Molecular dynamic simulations showed an interaction between miR-193a and other transcription factors like apolipoprotein 1 (APOL1), vitamin D receptor–retinoid X receptor (VDR-RXR) heterodimer, Yin Yang (YY) 1, and sex-determining region Y-Box2 (Sox2) [49]. Elevated miR-193a is directly associated with FSGS and crescentic glomerulonephritis in animal models. miR-193a is also noted to be elevated in studies of human FSGS, membranous nephropathy (MN) (presumed primary MN), and diabetic nephropathy (DN). Increased miR-193a expression causes podocyte cell death; in contrast, it is associated with the proliferation of PECs, as shown in crescentic glomerulonephritis [50]. Downregulation of miR-193a in cultured PECs promotes their transdifferentiation towards a podocyte phenotype [51,52]. MiR-193a is proposed as the master switch that decides the transdifferentiation of PECs to podocyte phenotype in vitro [51]. The modulation of miR-193a may be responsible for podocyte generation from PECs during growth and after injury [53]. PECs were shown to replace lost podocytes over the glomerular tuft in a mouse model of acute podocyte depletion. Labeled PECs were tracked from the onset of podocyte injury to day 28 and were found to migrate to glomerular tuft and express podocyte proteins increasingly with time, correlating with recovery [53]. Inhibition of miR-193a resulted in the resolution of crescentic lesions in an animal model of crescentic glomerulonephritis; however, the PEC switch to podocytes in this animal model was not studied [51]. The role of miR-193a in promoting PEC-to-podocyte transdifferentiation in human glomerular diseases is unknown, and the speculations are discussed in detail elsewhere [6]. The effects of miR-193a on various podocyte-relevant molecules and pathways are shown in Figure 2.

### 5.1. A. miR-193a in FSGS

FSGS is a histopathological diagnosis characterized by the glomeruli’s segmental scarring reflected as glomerulosclerosis resulting from podocyte (PD) injury. The loss of more than 20% of PDs results in the development of classical focal segmental glomerulosclerosis (FSGS) [59]. FSGS is considered primary or idiopathic if the cause is unknown, secondary if there is a known disease, and genetic in those with disease-causing mutations. Africans carrying APOL1 variants (G1 and G2) are prone to develop FSGS at several times higher than others with wild-type APOL1 [60]. The G1 variant is a compound missense mutation (S342G: I384M), encoding two nonsynonymous amino acids, while the G2 variant is a six bp inframe deletion resulting in the loss of two amino acids (N388 and Y389) at the C-terminal helix of APOL1. Approximately 34% of African Americans carry one of the two risk variants, and 13% have both coding variants [60]. Overt expression of APOL1G1 and G2 has been associated with PD injury in in vitro and in vivo studies [61,62].

Primary FSGS starts with podocyte injury and progresses to ESKD if not controlled. It is diagnosed after excluding known secondary causes of FSGS, such as genetic mutations, viral infections, drugs/toxins, and hemodynamic disorders. While no proven factor is identified as the cause of primary FSGS, several circulating permeability factors are proposed to be causative. Primary FSGS and secondary FSGS are typically not phenotypically different, except for the rapidity of onset of nephrotic syndrome, degree of foot process effacement, and presence or absence of glomerulomegaly, none of which are pathognomonic or specific [63]. Therefore, an unmet need exists in the differentiation of these two subtypes of FSGS.

MicroRNA-193a (miR-193a) is associated with FSGS, reflected by animal studies, human kidney biopsies, plasma, and urine [3,64,65]. Overexpression of miR-193a in podocytes is associated with podocyte injury, the rapid development of FSGS, and death in miR-193a transgenic mice [3]. miR-193a induction in transgenic mice led to the development of fatal FSGS within 6–12 weeks. The affected kidneys showed high expression of miR-193a in podocytes. The transgenic mouse model of miR-193a-induced FSGS is reproducible and a novel model with which to study FSGS further.

### 5.2. How Does miR-193a Cause Podocyte Injury?

miR-193a downregulates Wilm’s tumor protein-1 (WT-1) expression and the downstream effector pathways that maintain podocyte integrity. miR-193a binds to the coding sequence of the WT-1 gene to inhibit its expression. Glomeruli of miR-193a transgenic mice showed lower WT-1 levels in wild-type mice. These glomeruli were also noted to have lower gene expression of essential podocyte proteins like podocalyxin, nephrin, NOTCH-1, VEFG-A, podocin, and CD2AP. WT-1 mediates miR-193a’s regulation of podocyte phenotype and the expression of other podocyte-relevant genes. Others have noted increased gene expression associated with the WNT signaling cascade and extracellular matrix in podocytes of miR-193a-overexpressing mice [66].

### 5.3. Can miR-193a Differentiate Primary from Secondary FSGS?

Not all types of FSGS display glomerular overexpression of miR-193a, e.g., genetic causes of FSGS. Also, widely used animal models of FSGS, such as puromycin aminonucleoside and adriamycin nephrosis, were not found to show glomerular overexpression of miR-193a [67]. Mice injected with soluble urokinase receptor (suPAR) developed podocyte foot process effacement and showed no higher glomerular miR-193a expression in wild-type mice. miR-193a-expressing organs (other than kidneys) did not release circulating factors that caused FSGS in the experimental mouse model reported by Gebeshuber et al., as transplanted wild-type kidneys into the miR-193a transgenic mice were not affected by FSGS [3]. The study suggests that factors in the serum (similar to putative circulating permeability factors in primary FSGS) did not contribute to the development of FSGS in transgenic miR-193a mice. However, the possibility that circulating factors in patients with primary FSGS can increase miR-193a expression in kidneys cannot be discounted based on the study. Human kidney tissue with FSGS showed upregulation of miR-193a levels compared to normal kidneys or other glomerular diseases [3]. Children with primary FSGS (diagnosed by excluding secondary causes) had higher urinary exosomal miR-193a than those with other glomerular diseases, like minimal change disease and IgA nephropathy, and healthy controls [65]. Also, plasma miR-193a was higher in patients with FSGS than in the healthy controls [68]. Therefore, further studies evaluating the role of miR-193a in characterizing a specific subtype of FSGS are necessary. What causes miR-193a upregulation in FSGS? Which occurs first in the sequence of FSGS pathogenesis: increased miR-193a or podocyte injury? The answers remain elusive.

### 5.4. APOL1 and miR-193a

APOL1 is a component of high-density lipoprotein expressed widely in human organs/tissues besides in the plasma. Kidneys are an essential localization of APOL1 protein. APOL1 protein is observed to be expressed in podocytes (indirect immunofluorescence) and endothelial cells under normal conditions in all, irrespective of race/ethnicity. APOL1 is part of the podocytes’ cytoplasmic membrane and mitochondrial organelle. miR-193a has been proposed to alter APOL1 protein expression by post-transcriptional binding to APOL1 mRNA. High miR-193a reduces APOL1 (wild-type) levels in in vitro studies. miR-193a inhibition enhanced APOL1G0 expression and prevented podocyte dedifferentiation in cultured podocytes treated with high glucose [57]. Notably, APOL1G0 decreases miR-193a expression in in vitro studies [55]. Therefore, an inverse bidirectional relationship between APOL1 and miR-193a has been contemplated [55,56]. On the contrary, APOL1 risk variants (G1 and G2) enhance miR-193a expression in podocytes and cause injury [54,57].

### 5.5. miR-193a in Experimental Crescentic GN

In a mouse model of nephrotoxic nephritis induced by nephrotoxic antiserum, miR-193a overexpression was noted in the proliferating parietal epithelial cells (PECs) forming the “crescent” lesion [51]. Specific inhibition of miR-193a in this mouse model was associated with lesser severity of crescent lesions [51]. The investigators administered miR-193a inhibitor (locked nucleic acid (LNA)-193a-3p and 5p) one day after the nephrotoxic serum injection and euthanized the animals on day 10. Therefore, whether miR-193a inhibition prevents crescent progression or resolves an already formed crescent is unclear. The mechanism behind reduced proteinuria following crescent resolution was not explicit in the study description [51]. PEC transdifferentiation to podocytes upon inhibition of miR-193a may have caused proteinuria reduction. It could also be that miR-193a inhibition in the disease course prevented podocyte injury.

### 5.6. miR-193a in Diabetic Nephropathy

Almost 30–40% of diabetic patients develop a chronic kidney disease called diabetic nephropathy [69]. Pathologically, it is characterized by mesangial expansion loss of mesangial cells (MC) and podocytes (PD). Since MC proliferation and matrix accumulation are early features of diabetic nephropathy, MC injury was considered an initiating event. However, during the last decade, mechanistic studies revealed that the loss of podocytes (PD) in response to a high-glucose milieu (HG) is the primary event followed by mesangial expansion [70].

Elevation of miR-193a has been reported in the plasma of diabetic patients with DN [64] compared to those without DN. In a study by Gao et al. [26], WT-1 expression was reduced and miR-193a expression was increased in the glomeruli of 15 patients with DN compared to control kidney tissue. In diabetic mice who developed DN at 10 weeks of age, losartan, a renin-angiotensin-aldosterone system (RAAS) blocker, administered at 10 weeks of age for 12 weeks, reduced albuminuria and prevented decrease in expression in the glomeruli of mice. While the investigators reported a lowered miR-193a expression after losartan treatment in diabetic mice of 14 weeks of age compared to untreated diabetic mice, there was no increase in the baseline glomerular miR-193a in diabetic mice compared to control mice. Since these investigators did not analyze podocyte expression of miR-193a in diabetic mice, there is a possibility that diabetic mice might have a higher expression of miR-193a when compared to control mice. Another group of investigators [46] showed the reduced expression of WT-1 in mice with DN and cultured podocytes treated with high glucose. Podocytes were noted to have increased apoptosis, reflected by high BAX protein and low BCL-2 protein expression, besides having high miR-193a expression. Treatment with miR-193a inhibitor could prevent the downregulation of WT-1 expression. In turn, preserving WT1 could have protected apoptosis in the podocytes treated with high glucose. The exciting hypothesis that RAAS blockers work through miR-193a in causing podocyte stability warrants further experimentation. Notably, we earlier reported the effect of losartan on increasing vitamin D receptor (VDR) expression in cultured podocytes [58], the effect of vitamin D deficit on RAAS activation in cultured podocytes [71], and the effect of VDR agonist on reducing miR-193a expression in vitro [57]. Losartan may work on miR-193a via VDR upregulation. Also, losartan was shown to increase the number of renal progenitors and improve outcomes in a mouse model of mesangioproliferative glomerulonephritis [72].

### 5.7. miR-193a in Membranous Nephropathy

This is a common cause of nephrotic syndrome in the adult population. It is initiated by antibodies primarily directed to podocyte antigens such as phospholipase A2 (PLA2) receptor, thrombospondin type domain containing 7A (THSD7A), Nell-1, Exotocin-1/2, Sema3B, and PCDH-7 (primary MN) or associated with clinical conditions such as hepatitis B and systemic lupus erythematosus (secondary MN) [73]. Although pathology is not characterized by inflammation, immune system alteration is the dominant pathogenetic pathway in MN.

In 2019, two groups of investigators reported the association between miR-193a and MN in the Chinese population [27,74]. Zhang et al. observed high urinary exosomal miR-193a levels in patients with active MN [27]. Typical autoantibodies involved in primary MN were not reported; however, secondary causes of MN were excluded. Urinary miR-193a levels correlated with histopathological MN severity. In addition, urinary WT-1 and podocalyxin were noted to be low in these patients. The combination of high urinary miR-193a, a lower kidney expression of WT-1, and a lower urinary podocalyxin predicted poor kidney survival in these patients.

The other group, Li et al. [74], studied the effects of increasing and decreasing miR-193a in experimental MN rats and murine podocytes. MN was induced in Sprague-Dawley rats by injecting bovine serum albumin. MN rats were then injected with a miR-193a-NC and a miR-193a inhibitor. Control rats, MN rats, MN rats injected with miR-193a-nC, and MN rats injected with miR-193a inhibitor were compared for kidney expressions of miR-193a, WT-1, and other podocyte-relevant genes like podocalyxin, nephrin, and NOTCH-1. Apoptosis markers like caspase activity, Bax, and Bcl-2 expression were studied in kidney tissues. Light microscopy assessed kidney pathology in all these groups of rats after three weeks of the intervention. The MN rats and MN rats injected with miR-193a-NC had similar levels of miR-193a and WT-1 in the kidneys. Inhibition of miR-193a in MN rats (rats injected with miR-193a inhibitor) led to a reduction in miR-193a, an increase in WT-1 and downstream podocyte-relevant genes, and a reduction in apoptotic markers. Kidney tissues were shown to have no pathological changes in miR-193a inhibitor-injected MN rats. The timeline of events, such as the duration of MN before the intervention in rats, the time of rat kidney tissue analysis, and blood sampling in rats, could have been more straightforward. Also, miR-193a injection did not change the pathology of MN kidneys, which could be due to limited exposure before kidney tissue analysis. Further, the observation that inhibition of miR-193a could reverse kidney histopathology like podocyte structure, GBM thickening, and fibrosis in MN rats within three weeks of exposure is arguable.

Both these groups of investigators have implicated miR-193a in the pathogenesis of MN. Although MN and primary FSGS are glomerular disorders that share podocyte injury as a pathological feature, they are different entities in pathogenesis. Increased miR-193a expression in the kidney tissues of both these disorders implies that miR-193a upregulation could be associated with podocyte injury [3]. Nonetheless, lowering miR-193a improved kidney disease in both experimental animal models, suggesting a potential contributory role of miR-193a in podocyte injury in these models. Interestingly, the development of FSGS, not MN, in a miR-193a transgenic animal model indicates that miR-193a upregulation could be a secondary effect of immune-complex deposition in MN. However, enhanced levels of miR-193a contributed to additional podocyte injury, which the inhibition of miR-193a could have partially mitigated. Notably, the relationship between immune complexes and miR-193a is only a presumption, and it is not clear whether it is associated with specific immune complexes or a nonspecific response to any immune complexes.

An identical miRNA sequence, miR193b-3p, is also reported to be associated with kidney disease progression. miR-193b overexpression in noninvolved kidney tissues of patients with renal cell carcinoma was noted to be associated with a high risk of chronic kidney disease in the 12 months after radical nephrectomy [75]. Its expression also correlated with interstitial inflammation and fibrosis in these patients with no clinical evidence of kidney disease before nephrectomy.

## 6. Clinical Application of miR-193a in Glomerular Disease

Using miRNAs as biomarkers to diagnose glomerular diseases and using miRNA-based therapeutics is a promising approach for treating glomerular diseases (Figure 3). miRNAs can be measured in body fluids and tissues using quantitative polymerase chain reaction (PCR), next-generation sequencing, and microarray method [76].

### 6.1. MiR-193a as a Biomarker for Diagnosis and Prognosis of Glomerular Diseases

FSGS is diagnosed based on kidney biopsy, an invasive procedure. No single factor is known to cause primary FSGS; therefore, no biomarker has been employed to predict primary FSGS satisfactorily. The prognosis of FSGS is still assessed by response to glucocorticoids, which are associated with intolerable side effects. There is no reliable biomarker to prognosticate FSGS yet. Urinary exosomal miR-193a helps differentiate FSGS from minimal change disease in children [74]. Higher urinary miR-193a predicted faster progression of FSGS in children [65]. Higher levels of miR-193a, combined with the downregulation of WT-1 and podocalyxin, predict poor kidney survival in MN patients [27]. Similarly, an increased plasma miR-193a level predicted diabetic nephropathy in patients with diabetes and was associated with shortened kidney survival [64]. In a nutshell, miR-193a in urine and plasma seems to be a potentially helpful biomarker for FSGS diagnosis in children with nephrotic syndrome, for diabetic nephropathy diagnosis in patients with diabetes, and as a predictor of poor prognosis in FSGS, DN, and MN if validated by other investigators.

### 6.2. MiR-193a as a Therapeutic Agent for Glomerular Disease

miRNAs can be delivered in tissues using viral vectors. miRNAs can be inhibited by using chemically modified miRNAs such as oligonucleotides and locked nucleic acids [77]. Anti-miR-21 is being tested to retard the progression of Alport nephropathy as miR-21 inhibition in experimental models decreased fibrosis and inflammation in glomeruli and interstitium via its effect on PPAR/retinoid X receptor pathway [78]. While several miRNAs are being tested in preclinical studies exploring their impact on podocyte recovery [79] and glomerular diseases [80,81], none except anti-miR-21 have been tested in human clinical trials. The mode of delivery of miRNA inhibitor/mimic in human kidneys is unclear; if given systemically, the associated nonspecific side effects of miRNAs are potentially bothersome. On the contrary, oligonucleotide-based therapeutics like small interfering RNAs (siRNAs) targeting alternative complement pathway proteins are being tested in IgA nephropathy patients [81]. Developing and making oligonucleotide-based therapeutics is financially favorable compared to monoclonal antibodies. MiRNA-based therapeutics to treat glomerular diseases will likely see rapid growth in the near future.

## 7. Conclusions

MiR-193a is upregulated in glomerular diseases with podocyte injury in humans. Glomerular diseases like MN and FSGS are pathogenetically different; therefore, miR-193a upregulation is likely a secondary effect of the primary injury in these diseases. miR-193a is easily measured in human urine and serum besides kidney tissue, and the levels correlate with the severity of diseases like FSGS, MN, and DN. Inhibition of miR-193a improved glomerular lesions in animal models of FSGS, crescentic GN, and MN. A RAAS blocker, losartan, could reduce miR-193a in glomerular diseases directly or indirectly via VDR upregulation. Similarly, upregulation of APOL1G0 in the kidneys could reduce miR-193a levels and be exploited for clinical use. The clinical application of miR-193a inhibition in the therapy of glomerular diseases seems promising. Validation of miR-193a’s role in the diagnosis and prognosis of glomerular diseases by different groups of investigators across the globe is urgently needed. Some questions still unanswered in this subject are as follows: Is miR-193a the real culprit or a bystander of the pathogenetic process of glomerular diseases? Does miR-193a inhibition potentially prevent glomerular disease development, or does it induce the resolution of established glomerular disease? What could be an efficient way to administer miR-193a inhibitors and avoid side effects in human beings? How specific is miR-193a’s role as a biomarker to diagnose glomerular diseases like FSGS, DN, and MN? Appropriate hypothesis-driven studies will best address these questions.

## Figures and Tables

**Figure 1 biomolecules-13-01743-f001:**
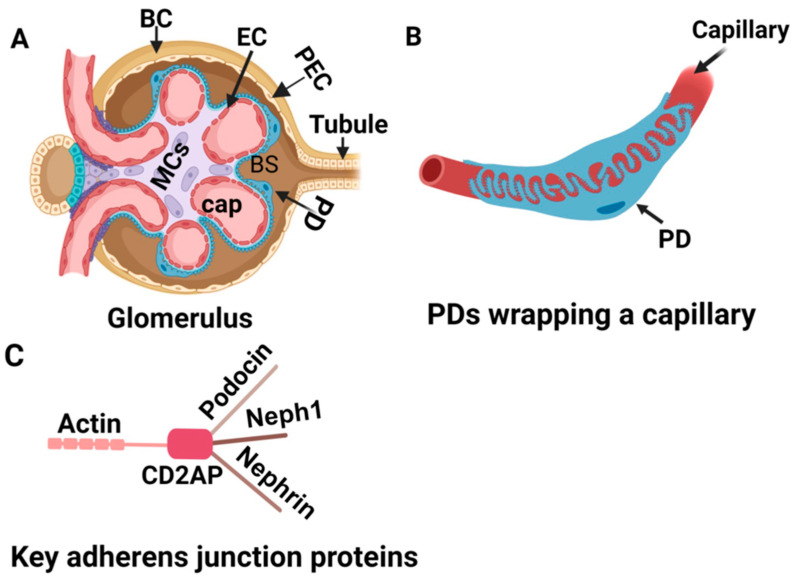
Schematic diagram showing a glomerular structure. (**A**) Glomerulus is a conglomeration of capillaries (Cap) starting from an afferent arteriole and ending in an efferent arteriole and residing in a Bowman’s capsule (BC). The outer wall of the Bowman’s capsule is lined by parietal epithelial cells (PECs), podocytes form the inner wall, and the filtrate passes through Bowman’s space (BC) to a tubule. Endothelial cells form the capillaries wrapped by podocytes (PD). Mesangial cells (MCs) and matrix remain in the center of capillaries composed of various cells. (**B**) A capillary is wrapped by podocytes. (**C**) Schematic representation of critical adherens junction proteins, which maintain the integrity of the slit diaphragm.

**Figure 2 biomolecules-13-01743-f002:**
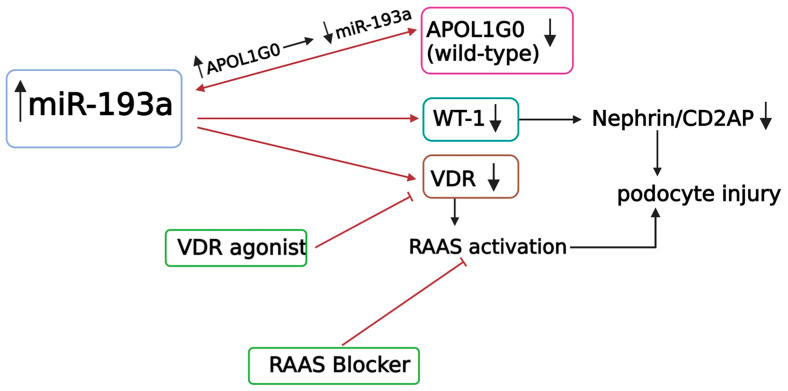
Effects of miR-193a on target messenger RNAs and its impact on podocytes [49,54,55]; miR-193a inhibits APOL1G0 [49,55,56]; miR-193a downregulates WT-1 [54] and VDR [49,57]. Attenuation of WT1 results in the downregulation of the expression of podocyte proteins (nephrin and CD2AP), which form adherens junction complexes and decreased expression of VDR, resulting in compromised heterodimerization of VDR with RXR [49] and activation of the RAAS [58]; RAAS blockers such as losartan increase VDR [58]. RAAS: renin-angiotensin-aldosterone system, VDR: vitamin D receptor, RXR: retinoid X receptor, APOL1G0: apolipoprotein wild-type.

**Figure 3 biomolecules-13-01743-f003:**
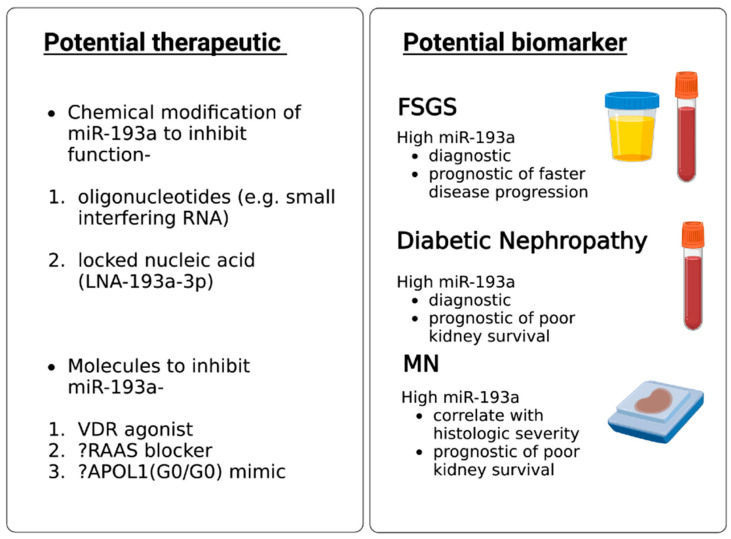
Potential clinical applications of miR-193a in glomerular diseases.

**Table 1 biomolecules-13-01743-t001:** MicroRNAs are involved in glomerular diseases (FSGS, DN, and MN).

MicroRNA	Mechanism of Action
miR-193a (FSGS, DN, MN) [3,26,27]	Targets Wilm’s tumor protein 1, disrupts podocytes.
miR-30 family (FSGS, DN) [19,28]	Modulates fibrosis and inflammation.
miR-29 family (FSGS, DN, MN) [29]	Regulates extracellular matrix.
miR-21 (FSGS, DN, MN) [30]	Promotes fibrosis and inflammation.
miR-135a (FSGS, DN) [31]	Inhibits transient receptor potential cation channel 1, alters calcium signaling.
miR-155 (FSGS, DN, MN) [32]	Contributes to podocyte injury.
miR-146a (FSGS, DN, MN) [33,34]	Regulates inflammation and fibrosis.
miR-93 (FSGS, DN, MN) [35,36]	Modulates podocyte dysfunction.
miR-25 (FSGS, DN) [36]	Impairs podocyte function and regulates pathways involved in DN progression.
miR-26a (FSGS, DN) [37]	Alters podocyte signaling pathways.
miR-181a (FSGS, DN) [38]	Modulates inflammatory response, targets tumor necrosis factor-alpha.
miR-378 (FSGS, DN) [39]	Modulates transforming growth factor-β1 signaling pathway.
miR-214 (FSGS, DN) [40]	Regulates fibrosis and inflammation.
miR-200 family (FSGS, DN, MN) [41]	Affects epithelial-to-mesenchymal transition.
miR-192 (FSGS, DN, MN) [42]	Modulates podocyte function.
miR-199a-3p (FSGS, DN) [43]	Affects podocyte integrity and protects tubular epithelial cells from high-glucose injury.
miR-204 (FSGS, MN, DN) [44]	Regulates podocyte injury.
miR-126 (FSGS, DN, MN) [45]	Regulates angiogenesis and inflammation.
miR-125b (DN, MN) [46]	Regulates inflammation and podocyte injury.
miR-143 (FSGS, DN) [47]	Affects podocyte structure and function.
miR-34a (MN, DN) [48]	Regulates podocyte apoptosis and fibrosis.

FSGS: focal segmental glomerulosclerosis, DN: diabetic nephropathy, MN: membranous nephropathy.

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
