# Peer review of "MicroRNA193a: An Emerging Mediator of Glomerular Diseases"

_biomolecules, 2023, doi:10.3390/biom13121743_

Round 1
Reviewer 1 Report
Comments and Suggestions for Authors
The authors Bharati et al. have presented a review on the role of MicroRNA193a as a mediator of glomerular disease. The authors present a comprehensive review on the current status of MicroRNA193a as a predictive biomarker for glomerular disease like FSGS, DKD and MN. However certain major concerns need to addressed:
1) The introduction is a bit confusing. While the set up for the role of miRNAs in glomerular disease is well written, the authors also present an alternate hypothesis that miRNA193a is involved in PEC to podocyte transition. However not much evidence is provided or expounded on in this regard. Discussing the role of PEC to podocyte transition in glomerular disease and speculating if miRNA193a is involved, would greatly benefit this review.
2) In section 4.3 the authors need to first tell readers what is the difference between primary and secondary FSGS. Are there specific phenotype or mechanistic characteristics that are specific to each type and then discuss how miRNA193a plays a role here. The authors also mention here that in classic models of FSGS (PAN and Adriamycin induced) miRNA193a levels in the glomerulus are unchanged. However, there is no further elaboration or discussion on this topic. Moreover they say "these data" prove that circulating miRNA193a does not play a role in mediating injury. How do the authors bridge these contrary statements.
3) Does miRNA193a have a physiological role? Since it closely regulates WT-1 expression, does it have a role in development of podocytes? Is it only upregulated in disease conditions or are there developmental effects as well?
4) The authors briefly touch upon a role for upregulated miRNA193a in cancer. Do the same mechanisms of toxicity and cell fate apply to glomerular disease. Or are the effects solely mediated by WT-1 mediated signaling.
5) A lot of references are missing within texts. The authors need to add in references where appropriate. The authors also need to include the references in the table in the bibliography (and add the corresponding numbers within the table).
Minor comments:
1) Do podocytes "line the inner wall of the Bowman's capsule" (line 37)?
2) I don't think podocytes "constitute the slit diaphragm protein" (line 38). SD is part of the podocyte architecture. The sentence needs to be rewritten.
3) Pre and Pri miRNA are used interchangeably.
4) The losartan study (Ref 46) only shows that treatment prevents a decrease in WT-1 expression in db/db mouse kidneys. It would be unfair to say that losartan increases WT-1 expression (line112) as the data does not support this.
Reviewer 2 Report
Comments and Suggestions for Authors
The manuscript focuses on the roles of MicroRNA193a in glomerular diseases. I have several comments that should be addressed.
1. The key content of the manuscript is not prominent. L30-L163 should be reworded and simplified.
2. According to Table 1, there are many miRNAs that are involved in glomerular diseases, why do the authors only revolve around MicroRNA193a in the following text?
3. In Table 1, the reference column should be replaced with a different representation instead of a website address.
Reviewer 3 Report
Comments and Suggestions for Authors
The review by Bharati et al. aims to describe the role of MiR-193a in glomerular diseases. The literature is up to date and the topic is well described. Nevertheless, there are some points that need to be fixed.
1. The Authors bring some concepts here and there, putting the reader on the spot making him go back and forth in the text. I suggest putting the description of the different glomerular diseases (lines 76-115) at the top of the respective paragraph with miR-193a (e.g., by putting lines 76-87 at the top of the paragraph beginning at line 61).
2. The title put miR-193 as the protagonist of this review but the Authors introduce miR-193 at page 8. I suggest the Authors to reduce the introduction paragraph (lines 30-51), then put the paragraph from line 116 to line 148. Here introduce glomerular diseases (lines 61-74) under the title of line 164.
3. APOL1 haplotypes would look better after line 93.
4. Paragraph from line 149 to line 163 would suit better as the beginning of point five (line 180).
5. Please bear in mind that FSGS is an histopathological sign not a disease and this is why you can find it in several pathologies. There is no mention of this in the text.
6. I suggest adding these papers: PMID: 32302353 and PMID: 36830635.
7. It could be useful also a paragraph briefly describing the role of miR-193a in renal cell carcinoma.
Minor observations
1. In my opinion, Figure 2 adds no information or clarification.
2. Legend to Figure 2 has a mistake, G1/G instead of G1/G2
3. Legend to Figure 3, line 33 APOL1GO instead of APOL1G0
4. Table 1 can be represented as a portrait with numbered references. In any case, references must not be reported as hyperlinks.
5. Also in legend to Figure 3 references must be reported as superscript numbers as per journal indications.
Round 2
Reviewer 1 Report
Comments and Suggestions for Authors
The authors have addressed all concerns previously raised.
Author Response
Thank you for your review.
Reviewer 2 Report
Comments and Suggestions for Authors
I have no further comments.
Author Response
Thank you for your review.
Reviewer 3 Report
Comments and Suggestions for Authors
The Authors improved the manuscript.
I've found only a couple of minor points:
-Figure 2 is cut
- Paragraphs 4.5, 4.6, and 4.7 have a typo since there are capital letter (B, C, and D respectively) with no meaning
Author Response
I have revised the manuscript as suggested by the reviewers and the revised version is enclosed.
